# iPSC-Derived Liver Organoids: A Journey from Drug Screening, to Disease Modeling, Arriving to Regenerative Medicine

**DOI:** 10.3390/ijms21176215

**Published:** 2020-08-27

**Authors:** Cristina Olgasi, Alessia Cucci, Antonia Follenzi

**Affiliations:** Department of Health Sciences, School of Medicine, University of Piemonte Orientale, 28100 Novara, Italy; cristina.olgasi@med.uniupo.it (C.O.); alessia.cucci@uniupo.it (A.C.)

**Keywords:** iPSCs, hepatocytes, endothelial cells, cholangiocytes, Kupffer cells, liver disease, liver organoids, liver bud

## Abstract

Liver transplantation is the most common treatment for patients suffering from liver failure that is caused by congenital diseases, infectious agents, and environmental factors. Despite a high rate of patient survival following transplantation, organ availability remains the key limiting factor. As such, research has focused on the transplantation of different cell types that are capable of repopulating and restoring liver function. The best cellular mix capable of engrafting and proliferating over the long-term, as well as the optimal immunosuppression regimens, remain to be clearly well-defined. Hence, alternative strategies in the field of regenerative medicine have been explored. Since the discovery of induced pluripotent stem cells (iPSC) that have the potential of differentiating into a broad spectrum of cell types, many studies have reported the achievement of iPSCs differentiation into liver cells, such as hepatocytes, cholangiocytes, endothelial cells, and Kupffer cells. In parallel, an increasing interest in the study of self-assemble or matrix-guided three-dimensional (3D) organoids have paved the way for functional bioartificial livers. In this review, we will focus on the recent breakthroughs in the development of iPSCs-based liver organoids and the major drawbacks and challenges that need to be overcome for the development of future applications.

## 1. Introduction

The liver is the largest organ of the human body that is responsible for several functions related to the maintenance of homeostasis. The liver works as an endocrine and exocrine gland covering essential body functions, such as bile production, plasma protein secretion, hormone synthesis, and drug metabolization. Liver functions are maintained by parenchymal cells, cholangiocytes, and hepatocytes, in direct contact with the blood through hepatic sinusoids, which are lined by liver sinusoidal endothelial cells (LSECs). Non-parenchymal cells are represented by LSECs and Kupffer cells (KCs), which are resident liver macrophages and stellate cells (Figure 1) [1].

An imbalance of liver function results in a pathological condition known as liver failure due to its fundamental role in the body. The causes can have genetic and/or environmental origin and lead to hepatitis, fibrosis, cirrhosis, cancer, metabolic, or autoimmune disorders. While liver transplants are the second most common form of solid organ transplantation, they only meet 10% of the global needs, with liver disease being the fifth-most common cause of death worldwide. Besides the high mortality rates, the lifetime treatment patients with liver insufficiency add to the financial burden of global health care systems [2,3].

Other approaches have been explored to treat liver failure, such as the xenotransplant, bio-artificial liver, and hepatocyte transplantation, due to the low availability of organs and a lifetime treatment of patients with immunosuppressive drugs [4]. Hepatocytes are commonly used because an appropriate number of cells can be isolated from livers that are not suitable for a liver transplantation. However, these treatments do not represent a definitive cure, but are commonly utilized as a temporary solution [5]. The major issue regarding hepatocytes is represented by a low survival and engraftment rate due to a lack of cellular communication between hepatocytes and the non-parenchymal cells. In fact, liver function is maintained by highly orchestrated interactions within the hepatic cellular network [6]. In consideration of these issues, further investigations are essential to ameliorate hepatocyte engraftment, including an optimal cell combination of adult hepatic cells (hepatocytes, LSECs, cholangiocytes, KCs, and stellate cells) to restore liver functionality. Hepatocytes, LSECs, and KCs are characterized by a low proliferation potential in vitro and it is difficult to maintain their phenotype. When considering these characteristics, induced pluripotent stem cells (iPSCs) represent an optimal alternative cell source. The iPSCs can be obtained through the ectopic expression of at least four transcription factors that are relevant for the maintenance of embryonic stem cell (ESCs) identity (f.i. Oct4, Sox-2, Klf4, and c-Myc), inducing pluripotency by reprogramming mature somatic cells [7,8,9]. The obtained iPSCs show an ESC-like phenotype with the potential of differentiating into several cell types, in addition to an unlimited self-renewal capacity. These cells are considered to be the leading candidate for a donor cell source in regenerative medicine [10]. As a result, several methods and transcription factor cocktails have been tested to generate bona fide and clinical grade iPSCs. However, despite this potential, two-dimensional (2D) cell cultures of iPSCs or primary cells are not enough to study cell-cell communications, tissue microenvironments, or replicate organ structures that mimic the in vivo environment. As a result, three-dimensional (3D) cultures were developed with the aim of replicating high cell-extracellular matrix (ECM) and cell–cell interactions [11,12]. They are referred to as organoids, which defines 3D aggregates that contain several cell types capable of self-organization. Organoids can be generated from iPSCs, ESCs or specific adult cells, making them a promising model for basic and translational research. Several attempts have been made to generate liver organoids by different methods, which results in vascularized organoids, cholangiocytes based organoids, and, in recent years, next generation liver models developed on a chip. All of these methods share the use of iPSC derived hepatocytes that are the best characterized from the perspective of differentiation. In this review, we will describe iPSC applications as a tool for differentiating into various liver cell types, including hepatocytes, cholangiocytes, endothelial cells, and KCs. Further, we will explore the advantages in generating self-assemble or matrix-guided 3D organoids to build functional bio-artificial livers that can be useful for disease modeling, drug screening, and regenerative medicine approaches for liver diseases.

## 2. iPSCs Applications

The use of iPSCs and iPSC-derived differentiated cells have been beneficial for the study of disease biology, in vitro disease modeling, and the development of new drugs, as well as a tool both for screening and toxicity tests. Most importantly, iPSC technology has the potential to be used in clinical applications. Because iPSCs can be differentiated into most cell types, they can be used as a model to study molecular mechanisms underlying disease development in targeted cell types. Such models allow for a better understanding of disease pathogenesis and the comparison between affected and healthy cells.

In this regard, iPSCs are a useful platform for drug discovery and development, thus reducing the high cost of generating new drugs [13]. The iPSCs, with their ability of self-renewal, can represent an unlimited source of cells for drug testing. The iPSC technology also allows for the screening of a library of human cell lines, which may represent the genetic and potentially epigenetic variations covering of a broad spectrum of the population [14], that can be useful both for preclinical drug discovery and the development of personalized medicine [15,16]. Besides their application in drug discovery, they can provide an accurate prediction of drug toxicity in humans being a vital element. For example, iPSCs can be differentiated into different cellular targets having a primary tissue-like phenotype and in an unlimited supply [17]. Variability in individual responses to potential therapeutic agents is another problem in effective drug development. The use of human iPSCs allows for the study of single nucleotide polymorphisms that are associated with the ability of an individual to metabolize and clear drugs and toxins.

Personalized medicine that is based on patient-specific iPSC generation, correction and differentiation is expensive and time consuming. On the other hand, cell therapy approaches are limited by immunological rejection. The main immunologic barrier to overcome between two individuals is the HLA. The HLA system is the most polymorphic locus in the human genome with almost 10,000 HLA-I and -II alleles [18]; therefore, matching donor cells and recipients for HLA is an endless task due to the countless possibilities. An alternative strategy is the generation of a bank of HLA homozygous iPSC lines and derived cells that could be transplanted in a short time and with minimal rejection in all individuals [19,20,21]. In studies that were performed in Japan, it was estimated that a database of 24,000 individuals would need to be examined to find at least one homozygous representative for each of the 50 different haplotypes [22]. Indeed, donors who are homozygous for an HLA haplotype could easily be identified from the HLA databases already in use in bone marrow donor records or cord blood banks [23]. In this way, HLA-haplotype banking of iPSCs would overcome the cost and immunological limitation of iPSC-based cell therapy, making iPSCs closer to a clinical application. In Japan, they are already at the stage of preparing iPSCs matches with the Japanese population at the major HLA loci [24]. An alternative approach to overcome immunological barriers is the generation of a “universal” donor iPSC line. Deuse et al. engineered iPSCs by overexpressing the CD47 gene and disrupting two HLA I and II genes. After differentiation into endothelial-like and cardiomyocyte-like cells, immunogenicity tests revealed no cytokines or antibody response when transplanted into allogenic humanized mice [25]. Similar approaches have been achieved for the generation of iPSC-derived HLA-universal platelets lacking HLA class I expression by knocking-out β2-Microglobulin gene [26,27]. Taken together, iPSCs can be considered an attractive tool for developing regenerative medicine applications.

## 3. iPSCs-Derived Liver Cells

Since the discovery of human iPSCs, several studies have been published demonstrating the achievement of liver-like cells differentiated in vitro (Figure 2). Differentiation protocols aim to resemble, as closely as possible, the early stages of liver development. During embryogenesis and, in particular, after the process of gastrulation, the posterior foregut endoderm gives rise to hepatic progenitor cells or hepatoblasts in a process called “specification”, in which signaling factors, such as fibroblast growth factor (FGF), bone morphogenetic proteins (BMP), hepatocyte growth factor (HGF), and Wnt play a key role [28,29]. Liver specification is the result of the convergence of the cardiogenic mesoderm and septum transversum mesenchyme (STM), in which also early endothelial progenitors play a relevant role [30]. At this stage, under the guide of hepatocyte nuclear factor 4 (HNF4), hepatoblasts proliferate and enter the STM to form the hepatic diverticulum known as the liver bud [31]. During the liver bud formation, the bi-potent hepatoblasts give rise to hepatocytes and cholangiocytes in a process that is finely regulated by transforming growth factor-beta (TGFβ), Notch, Wnt, BMP, and FGF [32]. After cell commitment, hepatocytes and cholangiocytes gradually acquire a mature phenotype that, alongside other non-parenchymal cells, gives rise to the definitive adult liver structure (Figure 1).

### 3.1. Hepatocytes

Hepatocytes are the parenchymal cells of the liver that represent 80% of the liver mass. They are specialized epithelial cells that are involved in crucial functions for the maintenance of physiological body homeostasis. Fully mature and functional hepatocytes are significant for basic research purposes and regenerative medicine, but mainly for cytotoxicity testing and drug development. The generation of hepatocytes from iPSCs cells can be obtained through defined culture conditions. Taking into consideration hepatocyte embryogenesis, most of the differentiation protocols combine growth factors and small molecules to replicate the stages of the definitive endoderm, hepatic specification, hepatoblast formation, and finally mature hepatocytes [33,34].

Hepatic differentiation protocols have been developed to first induce endoderm differentiation using Activin A, FGF2, and BMP4. This first step is followed by hepatoblast differentiation, by FGF2 and BMP4, and hepatic maturation with HGF and oncostatin M (OSM) [35,36,37]. The iPSCs-derived hepatocytes obtained with these protocols exhibit similar features to hepatocytes isolated from human livers, such as lipid and glycogen storage, plasma protein secretion, and urea synthesis [38,39,40].

Despite this, most iPSCs-derived hepatocytes are not fully differentiated and maintain some features of fetal or newborn hepatocytes [41,42]. In particular, the drug metabolism and detoxification ability of iPSCs-derived hepatocytes are not comparable to primary hepatocytes due to the lack of cytochrome P450 (CYP450) activity. However, this feature is typical for hepatocytes cultured in vitro and it is possible that the liver microenvironment can revert the CYP450 expression and resemble hepatic function [43]. The more recent protocols have relied on the use of decellularized scaffolds or extra cellular matrix components, such as laminins and collagens, to which iPSCs can attach in order to mimic the liver microenvironment during hepatocyte differentiation [44,45,46]. Moreover, other approaches have been explored using 3D scaffolds of synthetic origins, such as hydrogel or other nanofibers scaffolds, resulting in the generation of a more mature phenotype compared to 2D cultures systems, which are able to activate the expression of CYP450 and other mature liver markers [47,48]. Despite the promising results, 3D driven hepatocyte generation is impaired by the poor standardization protocols during scaffold manufacturing. Altogether, the most recent efforts in finding the best protocol for hepatocyte differentiation highlight a strong potential for in vitro studies as a good starting point for future clinical applications [49].

### 3.2. Cholangiocytes

Cholangiocytes are the epithelial cells lining the inner space of the bile duct tree. They are the main player of hepatocyte-derived bile modification through secretion and absorption processing [50]. Although hepatocytes and cholangiocytes derive from a common progenitor, the hepatoblast, cell fate depends on defined factors released from the microenvironment during embryogenesis. Several protocols have been established to differentiate hepatoblast progenitors to functional cholangiocytes by downregulating the signaling for hepatocyte commitment in order to obtain cholangiocytes from iPSCs. The composition of differentiating culture media pushes the commitment to cholangiocytes with epidermal growth factor (EGF), interleukin 6 (IL-6), sodium taurocholate, Jagged1, and TGFβ [51,52,53,54]. Induced-cholangiocytes show mature markers, such as SOX9, OPN, CK7, CK19, CFTR, and they are negative for the hepatocyte marker HNF4a.

### 3.3. Endothelial Cells

Endothelial cells (ECs) are epithelial cells of mesoderm origin that line the inner layers of the blood and lymphatic vessels. Besides the role of being a vessel barrier, ECs are actively implicated in the immune response, inflammation, and transportation processes [55].

Depending on the location, it is possible to distinguish different subtypes of ECs according to their morphology, vasculogenesis, or angiogenesis in relation to organ-specific functions. Morphologically, they can be characterized as continuous, fenestrated or sinusoidal, depending on their function [56]. For example, liver sinusoids display a discontinuous endothelium that allow the exchange of solutes and macromolecules, while the endothelium of large vessels display tight junctions that serve as a barrier [57]. Subtypes of ECs can be distinguished according to their origin whether it be arterial, venous, or lymphatic, with their size determining whether they are macrovascular or microvascular ECs. Several markers have been found to be associated with these subtypes of endothelial cells: venous subtypes express Notch4, ephrin type-B receptor 4, and Coup-transcription factor II, while arterial subtypes express EphrinB2 and Notch1, with lymphatic subtypes expressing podoplanin, prospero homeobox protein 1, and lymphatic vessel endothelial hyaluronan receptor 1 [57,58]. Subtypes of ECs can be identified by these unique markers; however, some common markers are shared among all ECs, such as CD31, vWF, CD144, and VEGFR2 [59]. Primary ECs can be obtained from endothelial biopsies or peripheral blood but they are difficult to retrieve and have a limited proliferation potential [60]. Moreover, given their potential therapeutic applications, it has been necessary to generate scalable quantities of ECs. For this purpose, iPSCs are an attractive source for generating ECs based on their self-renewal potential. Several methods have been developed to differentiate iPSCs into ECs and it is possible to summarize them into four approaches: co-culture, embryoid body (EB) formation, and 2D and 3D culture. Each of these methods are based on the evidence that vascular ECs are derived from a common precursor, the angioblast, which differentiates into ECs in the presence of VEGF mediated by TGFβ2 [61]. The co-culture approach involves iPSCs grown on murine bone marrow stromal cells (OP9 cells), but the efficiency of the differentiation process is low, and the use of murine feeder layers is not suitable for translation to the clinic [62,63,64]. The use of the EB formation method is based on the possibility of obtaining all three germ layers from EBs reproducing the process of embryogenesis. Several studies have, in fact, demonstrated that it is possible to obtain the cell of interest by adding specific cytokines to EB medium. For EC differentiation, BMP4 has a crucial role in differentiating iPSCs into mesoderm commitment [65,66,67,68]. To this, it can be added VEGF at different concentrations or different forms, based on the type of ECs to be obtained [69]. For example, high concentrations of VEGF can induce iPSCs to differentiate into arterial subtypes, whereas VEGF-C and angiopoietin 1 can favor lymphatic differentiation [57]. Aside from these cytokines that are commonly used in the EB method, there are others that are involved in differentiation, such as Activin A and bFGF [70]. Additionally, it has been demonstrated that the suppression of TGFβ pathway during specification and maintenance can increase the yield of ECs, thus promoting the ECs lifespan [71]. Although being an improvement from the co-culture method, the EB method is not the best one for achieving a high yield of ECs. For this reason, 2D and 3D methods have been developed. The 2D approach involves the use of matrixes, such as collagen, matrigel, fibronectin, or gelatin, and a two-step protocol where BMP4, activin A, bFGF, and VEGF are first used to induce mesoderm specification. Subsequently, the resulting cells are treated with Y27632 (inhibitor of associated protein kinase), SB431542 (inhibitor of TGFβ), VEGF, and bFGF to induce the final differentiation step and promote proliferation [72]. Moreover, by modulating the cytokine concentrations, it is possible to guide the differentiation process towards a specific lineage. For example, Palpant et al. demonstrated that by modulating BMP4 and activin A concentrations, it was possible to obtain hemogenic and cardiomiogenic mesodermal precursors [73]. This 2D differentiation method achieved a high yield of mature cells and decreased the time of the overall process. Finally, a 3D protocol was developed to obtain a more efficiently ECs by using different scaffolds, where iPSCs can be differentiated. For example, Zhang et al. used a fibrin scaffold and obtained ECs with a 45% efficiency. In general vessel specificity of iPSCs-derived ECs is difficult to obtain and, therefore, more specific protocols were investigated [74]. Ditadi et al. demonstrated that CD34^+^CD73^med^CD184^+^ cells can become arterial ECs, while, from CD34^+^CD73^+^CD184^−^, venous ECs can be obtained. VEGF-A and NOTCH were identified as specific factors driving arterial specification. Additionally, shear stress or hypoxia can increase the yield of arterial ECs from iPSCs-derived ECs [75].

It has been demonstrated that iPSCs-derived ECs can be used to model cadmium-induced atherosclerosis, in order to study coronary artery disease in the context of diabetes both to investigate endothelium dysfunction during disease or as a tool to induce vascularization and promote survival of islet cells in regenerative medicine approaches [76]. The iPSC-ECs can promote organogenesis by enhancing the functionality of generated tissues. Recently, iPSCs derived brain microvascular ECs were able to support the specification of neuronal cells into spinal cord neural tissue with an “organ on a chip approach” [77]. A similar approach was achieved in the formation of 3D myocardial tissue by combining human iPSCs-derived cardiomyocytes, vascular mural cells, and ECs showing a well-defined structure and functional electromechanical properties [78].

Despite numerous studies published regarding the differentiation of ECs, only one group has achieved the differentiation of iPSC into LSECs. Koui et al. obtained LSEC progenitors after mesoderm induction by inactivating the TGFβ pathway using the inhibitor A83-01 under hypoxic culture condition. After CD31, CD34, and FCGR2 positive enrichment, LSECs express specific markers, such as FLK1, F8, STAB2, and LYVE1, exhibiting a mature endothelial morphology [79]. Recently, the same protocol for LSEC differentiation was confirmed by Danoy et al., who further characterized the generated cells with additional specific markers, including CD144 and STAB1, and a new set of genes overlapping the gene expression profile of primary human LSECs [80]. When considering the contribution of ECs and iPSCs-derived ECs in promoting organogenesis, their role needs to be further investigated to improve these promising approaches.

### 3.4. Kupffer Cells

The Kupffer cells (KCs) are liver resident macrophages and they represent the 10% of the hepatic non-parenchymal cells, while being the largest population of resident macrophages in the body [81]. They adhere to sinusoidal endothelial cells supporting their ability to regulate hepatic function. The KCs can have a tolerogenic function preventing immune responses to immune reactive molecules present in the hepatic sinusoid [82]. However, in the case of inflammatory disease, they can shift to an activation state causing hepatocellular damage [83,84,85]. In an inflammatory state, KCs can release pro-inflammatory cytokines, such as tumor necrosis factor alpha (TNFα) and IL-6, growth factors, and reactive oxygen species [86,87]. In co-transplantation experiments, KCs allow liver regeneration after ischemic injury producing VEGF, thus being involved in both structural regeneration and neovascularization [88]. The use of KCs in co-culture studies with hepatocytes or in the generation of liver buds do have their limitations, with the loss of their functionality after isolation and the complexity of maintaining them over long-term [89,90]. To overcome these issues, several protocols have been developed for tissue macrophage differentiation from iPSCs, despite a limited information regarding KC differentiation [91,92,93,94]. The difficulty in generating KCs from iPSCs is derived from their embryonic development. It has been demonstrated that KCs originate from primitive macrophages generated in the yolk sac from early erythroid-myeloid progenitors [95,96,97]. These progenitors under hepatic signals can differentiate into liver-specific macrophages independently of the transcription factor Myb [98,99]. Recently, Tasnim et al. generated iPSCs-derived KCs starting from macrophage precursors and adding a hepatic stimulus [100]. Embryoid bodies were formed and they were cultured in the presence BMP-4, VEGF, SCF, ROCK Inhibitor, M-CSF, and IL-3, with each added at different time points to obtain macrophage precursors [92]. Pre-macrophages were cultured in a human hepatocyte conditioned media, as hepatic stimulus, in order to obtain KC expressing specific markers, such as CLEC-4F, ID1 and ID3, and acquiring phagocytosis properties, typical of KCs, and the ability to secrete IL-6 and TNFα upon stimulation [86]. Thus, this process represents a useful in vitro model for liver bud generation and the study of liver diseases.

## 4. Generation of Organoids

In recent years, it has been widely demonstrated the ability to isolate and maintain in culture primary cells of animal and human origin. Two-dimensional (2D) cell cultures were fundamental for cell biologists to study molecular mechanisms and preclinical testing. However, it emerged the importance of studying cell-cell communications, tissue microenvironment, and to replicate organ structures to mimic the in vivo environment. This led to the development of three-dimensional (3D) cultures with high cell-extracellular matrix (ECM) and cell-cell interactions [11,12]. They are referred to as organoids, which define the 3D aggregates containing several cell types that are capable of self-organization [101]. Morphologically, organoids show spherical shapes and can be maintained in suspension or embedded in different matrices [102]. The main difference between 2D and 3D cultures are the different exposures to signaling molecules and nutrients. In 2D cultures, the cells are disposed as a monolayer, thus all cells have access to the stimuli. On the other hand, in 3D culture, the cells at the center of the organoid are less exposed to the factors, resembling more the physiological microenvironment and providing a model of in vivo biology. The 3D cultures offer a better approach for drug testing, and they explain the failure of drug screening in 2D culturing system [103]. As such, organoids represent an innovative tool for embryogenesis, disease modeling, toxicology, drug screening, and transplantation purposes. Organoids can be generated from iPSCs, ESCs, or specific adult cells, such as epithelial cells or epithelial and mesenchymal cells, in the presence or absence of matrixes. The first reference to organoids come from a culture obtained from the dissociation of cells derived from iPSCs or ESCs that are capable of self-organization [104]. The first studies on re-aggregation were performed using epithelial cells. Bissel et al. demonstrated that primary cells from mouse mammary glands, embedded in an extracellular matrix (hydrogel) could reorganize into glandular structures [105,106,107]. Several studies subsequently demonstrated that an ECM hydrogel could replicate cell-to-cell and cell-to-ECM interactions. Using this method, it was possible to co-culture epithelial cells and fibroblasts and induce the differentiation of several types of gastric cells [108]. Even though the advantages of this new culture method could lead to the development of tissues, the limitation was represented by the short-term culture period. In 2009, Sato et al. developed a 3D system to culture Lgr5+ stem cells that were isolated from mouse intestinal tissue in ECM supplemented with R-spondin1, an LGR4/5/6 ligand that upregulates Wnt signaling, helping the maintenance of stem cell populations [109]. This organoid survived in vitro for more than three months and led to the use of isolated Lgr5+ stem cells as a cellular source for organoid cultures. Consequently, with these results, it was possible to generate human intestinal organoids. Moreover, using R-spondin1 in the culture media of Lgr5+ progenitor cells, established a starting point for the generation of organoids from other organs, such as colon, stomach, and liver [110].

## 5. iPSCs-Derived Liver Organoids

The first attempt of in vitro hepatic structure was reported by Michalopoulos et al., providing the initial information regarding the specific signals that are needed for assembling hepatic tissue in culture. Adult rat hepatocytes were isolated and maintained in roller bottles coated with collagen type I in a medium containing dexamethasone, HGF, and EGF, and, after 18–20 days, the cells organized themselves into a sheet with a typical hepatic configuration. It was demonstrated that HGF and EGF, are essential factors for the development of hepatic tissue and dexamethasone is necessary for hepatocyte maturation. However, the obtained hepatic sheets only survived in culture for a short period of time [111]. Subsequently, a long–term culture of self-renewing organoids was obtained from mice Lgr5+ cells that were isolated from damaged livers in the presence of R-spondin1 [112]. Indeed, only by adding R-spondin1 to the culture medium, cells exhibited self-renewing properties maintaining the ability to expand long-term as adult ductal progenitor cells, and to differentiate into hepatocytes. In recent years, several studies have focused on different methods to obtain liver organoids with the aim of mimicking the hepatic function. These methods can be summarized into four groups: co-culture methods, iPSC-derived organoids, liver on a chip, and bio printing technology.

### 5.1. Co-Culture Methods

The first attempt to generate a 3D liver was established by Takebe et al. by co-culturing iPSCs-derived hepatic endoderm like cells with mesenchymal cells and human umbilical vein endothelial cells (Figure 3A and Table 1). They were able to obtain vascularized liver bud organoids that, after transplantation in mice, gave rise to mature hepatocytes [113]. This was a demonstration that the formation of functional vessels could be the trigger to stimulate the maturation of the iPSCs-derived liver buds [114]. Asai et al. showed that paracrine factors secreted by mesenchymal cells and ECs, such as HGF [37,115,116], ANG, A2M, and PLG [117,118,119,120], could induce the formation of liver organoids, with hepatic differentiation and maturation evident once transplanted in immunodeficient mice, with a relevant albumin secretion [121]. It was also highlighted that hypoxia inducible factor (HIF) can interfere with albumin secretion and hepatic differentiation, further confirming the importance of LSECs as the starting point in liver bud organization and regeneration [122]. Other approaches attempted the generation of liver organoids entirely from iPSCs-derived cells (Table 1). Koui et al. differentiated LSECs and hepatic stem cells from iPSCs by modulating TGFβ and Rho signaling pathways, which are known to respectively regulate the proliferation and maturation of LSECs and HSC progenitors [79,123]. Obtained iPSCs, were further shown to have self-renewal properties in 2D culture systems, making this new protocol instrumental for liver modeling in vitro. An alternative approach has recently been described generating hepatobiliary organoids from iPSCs using Activin A, BMP4, BMP2, FGF4, HGF, OSM, and dexamethasone in a 45-day differentiation protocol. The authors developed a three-stage protocol in order to commit first both mesoderm and endoderm and later both hepatic and biliary co-differentiation inducing the maturation of hepatobiliary organoids that, once transplanted in immunodeficient mice, were able to survive up to eight weeks [124]. This model is useful to study liver development and can be applied in regenerative medicine approaches for liver diseases. Pettinato et al. developed an alternative protocol for liver organoid generation starting from human embryoid body formation. Specifically, human adipose microvascular endothelial cells were mixed with iPSCs and cultured in EB plates in a specific medium for hepatic differentiation containing Activin A, TGF-β, FGF4, BMP4, HGF, and Wnt pathway inhibitors. This new protocol led to an elevated hepatic differentiation and persistent albumin secretion in vivo [125].

### 5.2. iPSC-Derived Organoids

Besides co-culture protocols, only a few studies have focused on the development of liver buds starting from a homogenous cell population to directly differentiate iPSCs into organoids (Figure 3B). Guan et al. induced iPSCs to form hepatoblast aggregates that, after dissociation and culture in Matrigel gave rise to organoids containing both hepatocyte and cholangiocytes [126]. In brief, they cultured dissociated iPSCs in a medium containing BMP4, FGF2, with inhibitors of the Wnt and the PI3K pathways to induce endoderm differentiation and foregut spheroid generation. The spheroids were further cultured in a low concentration Matrigel scaffold in the presence of FGF10, HGF, OSM, and dexamethasone to induce the formation of 3D structures. Finally, Ouchi et al. initially differentiated iPSCs into foregut spheroids that, in the presence of an hepatocyte specific medium, was able to obtain definitive liver organoids containing all liver cell types and exhibiting a transcriptomic profile comparable to hepatic tissue [127].

### 5.3. Liver-on-a-Chip

Liver organoid technology based on 3D cultures provide models for liver development, tissue biology, and pathology mimicking liver architecture and improving cell-to-cell and cell-to-matrix interactions (Table 1). However, the challenge remains to build up on the liver complexity; thus, new technologies, such as microfluidic and liver-on-chips, are rapidly developing [137] (Figure 3C). These methods can be matrix independent or dependent or can be based on 3D bioprinting. The matrix independent technology uses a hanging drop platform, perfusion chambers, or microwell arrays that facilitate 3D spheroid organization and liver specification of EBs or primary cells [138,139,140,141,142]. In matrix dependent 3D culture, several ECM components are used to mimic the native liver microenvironment. The most commonly used ECM matrix are collagen [128], Matrigel [129], and hydrogels [130] to coat microfluidic chips aiming at the formation of 3D aggregates by providing native shear stress forces.

### 5.4. 3D Printing Technology

Currently, the 3D printing technology is rapidly developing to facilitate the fabrication of the 3D architecture and especially to study liver diseases and drug screening (Table 1) [131,132,133,134,135]. Generally, gelatin-methacryloyl (GelMA) Hydrogels are used as ink where primary hepatic cells are included and printed in transwell or perfused microwells to induce the generation of a 3D liver-on-a-chip. Recently, bio-compatible ink has been developed by using alginate, a calcium chloride solution and pluronic polymer used to print iPSCs-derived parenchymal and non-parenchymal cells to generate 3D liver organoids showing hepatic functions in vitro [136] (Figure 3D). Even though liver-on-a chip techniques offer advantages in generating functional liver organoids, it remains a complex and expensive technique that requires further study. 

## 6. Liver Organoids Applications

The self-renewing properties of organoids and the possibility to expand iPSCs and cells differentiated from iPSCs make organoids a promising model for basic and translational research. They have been used for the study of stem cell behavior, liver organogenesis, disease modelling, drug screening, toxicology, and regenerative medicine. Moreover, it is possible to generate biobanks of healthy or diseased human organoids making this technology a significant source for future studies [143].

### 6.1. Regenerative Medicine

Currently, liver transplantation is the only effective treatment for liver failure, with organ availability limited and patients needing lifelong immunosuppressive therapy. With the need of alternative approaches, the xenotransplant, bio-artificial liver, and hepatocyte transplantation [4] were developed; however, these treatments do not represent a definitive cure and they are commonly used as a temporary solution while waiting for liver transplantation [5]. As such, the generation of liver organoids can represent a suitable alternative for the treatment of hepatic diseases (Table 2). The first evidence of the applicability of liver organoids goes back to 2013, when Huch et al. generated bile-duct derived organoids, from Lgr5+ stem cells, which differentiated into functional hepatocytes after transplantation into Balb/c nude mice [112,144]. Moreover, after acute liver damage, bile-duct derived organoids differentiated into hepatocytes that are able to engraft and proliferate in the liver of transplanted mice. Finally, using the same differentiation protocol, they were able to generate organoids from A1AT-deficient patients that could be used for disease modeling. Recently, hepatic organoids generated from primary hepatocytes of both mouse and human origin, once transplanted in the damaged liver of immunodeficient mice, engrafted and survived for up to 90 days exhibiting an 80% engraftment [145]. Rashidi et al. generated an organoid from iPSCs-derived hepatocytes that, after transplantation in a polycaprolactone scaffold, a successful engraftment was demonstrated with detectable serum human albumin demonstrated in two murine models of tyrosinemia [146]. Finally, a good manufacturing practice (cGMP) compliant method was established in order to generate iPSCs-derived hepatocytes that, when seeded on a 3D poly-ethylene glycol-diacrylate scaffold were able to fully differentiate into hepatic tissue structures [147]. The transplantation of alginate encapsulated organoids in the peritoneal cavity of immunocompetent mice, were able to survived and secreted human albumin in a short-term experiment, without an immune reaction. These studies can be considered as a proof of concept for the applicability of liver organoids in future clinical approaches.

### 6.2. Drug Screening and Toxicity Test

The use of iPSCs as a tool for drug screening to evaluate their potential, toxicity, and predict the effects of candidate drugs, has been reported extensively [148,149]. However, it is known that the patient’s genetic background can influence the effects of the drugs being tested imparting a different response. iPSCs-derived organoids in 2D culture disposed as a monolayer do not represent the physiological microenvironment, because all cells have access to the stimuli. This can explain the failure of drug screening in some cases where 2D cultures have been used [103]. For this reason, organoids are emerging as an innovative tool for drug screening and toxicology testing. In 3D cultures, cells at the center of the organoid are less exposed to the factors, thus resemble the physiological microenvironment and provide a model of in vivo biology that can also be used for personalized medicine [104]. In this context high throughput screening of small molecule libraries have been used to develop drugs for liver diseases and test possible toxic effects by using iPSCs-derived hepatocytes as a model [150,151,152]. However, it emerged the importance of generating organoids composed of several liver cell types to uncover drug side effects based on the potential role of non-parenchymal cells in hepatotoxicity. For example, KCs have an important role in drug-induced liver injury and, for this reason, hepatocytes were co-cultured with KCs to achieve a more sensitive cellular model [100]. In recent years, liver organoids were also used to test tumor sensitivity to drugs. Broutier et al. established organoids from hepatocellular carcinoma, cholangiocarcinoma, and hepatocellular-cholangiocarcinoma, thus replicating the architecture and expression profile of the parental tumors [153]. These organoids can open opportunities for drug testing, approaches for personalized medicine and can be used for the generation of tumor bio-banks that are useful as screening platforms [154]. Finally, “liver-on-a-chip” can represent a platform for drug development and toxicology tests also allowing for pharmacokinetic and pharmacodynamic studies. Wand et al. generated liver organoids, using a perfusable microcapillary chip, displaying hepatocyte and cholangiocytes. The obtained organoids allowed for the measurement of a dose and time-dependent hepatotoxic effect of acetaminophen supporting the idea that “organ-on-a-chip” can represent a novel and valid platform for drug testing [155]. Even though liver organoids are an evolution of the 2D culture, drug induced liver injury still needs further investigations.

### 6.3. Disease Modeling

Alongside the application of 3D liver organoids for regenerative medicine and drug discovery purposes, disease modeling constitutes an appealing approach to understand the molecular pathways that are involved in the pathogenesis of liver diseases (Table 3). Animal models of disease still represent the best approach for biomedical research, but the possibility to use patient-derived-iPSC organoids could improve our knowledge regarding the mechanisms that lead to the unique disease phenotype in humans. Moreover, after the discovery of an “easy-to-use” gene editing platform as CRISPR/Cas9, it is possible to introduce or revert specific mutations to replicate the physiopathology in liver organoids [156]. In 2015, two groups modeled cystic fibrosis in cholangiocytes organoids from patient-iPSCs. Mutations in cystic fibrosis transmembrane conductance regulator gene (CFTR) impair chloride ion channels that can lead, at the liver level, to the blockage of intrahepatic bile ducts. In these studies, organoids were used which presented non-functional CFTR proteins with the subsequent impairment of chloride channel and inability to cyst formation, resembling the disease phenotype. In this model, the effects of the drug VX809 were tested, demonstrating the functional rescue of the mutated CFTR protein [52,157].

Liver organoids were also generated from iPSCs of patients affected by Alagille syndrome (ALGS), a multi organ disease that is caused by mutations in the JAG1 gene involved in biliary ducts formation. The JAG1 mutated iPSCs did not form tubular structures, impairing organoid generation. Interestingly, the JAG1 gene was disrupted by the CRISPR/Cas9 approach in healthy iPSCs, obtaining a similar impaired phenotype as compared to the ALGS patient-iPSCs [126]. Akbani et al. used patient-specific iPSCs to model citrullinemia type 1, an inherited metabolic disorder caused by mutations in the argininosuccinate synthase 1 (ASS1) gene. Liver organoids generated from endodermal EpCAM^+^ cells resemble the phenotype of the disease defined by hyperammonemia and decreased urea production. By lentiviral vector transduction of ASS1 wild-type gene in specific-patient organoids, the disease phenotype was partially rescued, demonstrating that organoids can be genetically manipulated efficiently [158].

In the context of acquired liver diseases, Nie et al. proposed studying hepatitis B virus (HBV) infection susceptibility in iPSC-derived liver organoids. Despite the fact that several models have been established for HBV infection, they do not resemble all genetic backgrounds [159]. When compared with 2D standard hepatocyte cultures, the self-organized liver organoids proved to be more susceptible to HBV infection. Moreover, the infection led to a down-regulation of specific hepatocyte markers and a high level of transaminases demonstrating a reduced hepatocyte function and altered morphology. Overall, these findings indicate that liver organoids could be used as a good HBV infection model for reproducing the virus life cycle and understand how individual genetic background can affect the pathogenesis of the disease [160].

### 6.4. Liver Cancer Organoids

Liver cancer is the most prevalent malignancy and the incidence has increased in the last years [161]. Hepatic cancer can be distinguished into primary or secondary malignances. Primary hepatic tumors are mainly hepatocellular carcinoma, cholangiocarcinoma, and mixed liver cancer. The main difficulty in finding a cure for these pathologies is due to their heterogenicity. It has been demonstrated that multi-gene mutations mainly cause liver cancer, where often within the same tumor the cancer cells do not show the same mutations [162,163]. On this basis, the development of personalized treatments is essential. Tumor liver organoids, also called tumoroids, can resemble the progression of the malignancy and can be used not only for drug screening, but also to study cancer development and define the precise treatment [164]. Tumoroids can be generated from needle biopsies, from patient-derived xenografts (PDX) or from iPSCs (Table 4). Takai et al. demonstrated that hepatocellular carcinoma (HCC) cells isolated by needle biopsy, could be cultured in porous alginate scaffolds. The generation of HCC spheroids mimic numerous features of glandular epithelium, such as the expression of EpCAM. Importantly, EpCAM positive HCC cells showed tumorigenic and metastatic potential in vivo and could be used as a tumor model [165]. The advantage in the use of needle biopsies is represented by the possibility of obtaining samples without a liver resection and they can further facilitate the generation of biobanks of liver tumoroids. The maintenance of tumor cells in vitro, however, does not replicate the microenvironment interactions, especially vascularization, interactions with the stroma, and the immune system [166]. Therefore, murine xenograft models were established to enable the implantation of human cancer cells. After tumor growth, they can be cryopreserved, expanded or used for drug testing. Most HCCs are insensitive to conventional chemotherapy due to the multi gene mutations. Broutier et al., using PDX organoids, performed a drug sensitivity experiment showing that ERK inhibition could have an effect on the progression of hepatic carcinomas [153]. Gu et al. generated a cohort of 65 liver cancers from PDX models and tested a multi-kinase inhibitor that targets FGFR1 and showed a therapeutic effect [167]. This drug was so effective that it was then used as a treatment for patients with advanced HCC [168]. Recently, a dataset of PDX has been developed from 116 HCC containing information regarding the expression profiles and the genetic alterations, which may led to the identification of biomarkers for personalized medicine [169]. PDX organoids can also be used as preclinical models mimicking the structure and the genetics of cholangiocarcinoma tumors. For example, is was demonstrated that cholangiocarcinoma can also be derived from differentiated hepatocytes [170]. Thus, by culturing cholangiocarcinoma organoids, Saito et al. were able to restore some hepatic function [171]. However, all these PDX organoids were generated from immunocompromised mice and are not suitable for immunotherapeutic approaches. To study cancer immunotherapy, it is fundamental to reproduce the human immune system and one strategy is the use of humanized mouse models for the implantation of human liver tumor fragments [172,173]. In a pivotal study, human HCC PDXs were generated in NSG gamma null mice that were repopulated with adoptive chimeric antigen receptor (CAR) T cells [174]. The arising PDX organoids showed the transcriptional, morphological, and immunological characteristics of the primary tumor and CAR T cells directed against a HCC tumor-associated antigen were able to suppress tumor growth [175]. More recently, human HCCs were generated in NSG mice with HLA-matched human immune systems and the organoid models were responsive to immunotherapies [176]. However, the humanized mice should be generated with the same immune system of the corresponding PDX, but the repopulation of the mice with hematopoietic stem cells from cancer patients was not found to be optimal [173]. Nevertheless, this system represents an important step for the development of a personalized and humanized mouse model for liver cancer research. To model cancer, organoids derived from healthy iPSCs or normal tissues can be used, and cancer gene mutations can be induced by CRISPR/Cas9 system. In particular, a recent study has demonstrated that the combination of BAP1 loss-of-function mutation and cholangiocarcinoma mutations (TP53, PTEN, SMAD4, and NF1), induced by CRISPR/Cas9 in normal liver organoids, can affect epithelial tissue organization and cell-to-cell junctions, resulting in the acquisition of malignant features [177]. Therefore, liver cancer organoids can mimic cell-to-cell and cell-to-ECM interactions and the drug sensitivity of human cancers and they can be useful for the development of personalized cures.

## 7. Challenges and Limitations of Liver Organoids

Liver organoids, an advancement of standard 2D culture system, is an elite platform to understand early steps of liver embryogenesis and a tool to explore applications that range from disease modeling to drug screening, envisioning approaches for regenerative medicine. In contrast to 2D cell culture, organoids contain several cell types that can self-assemble into 3D aggregates replicating cell-cell interactions that allow a selective exposure to signaling molecules and nutrients that mimic the liver environment. The discovery of iPSCs boosted the organoid technology by overcoming issues that are related to short-term availability of primary cells. On the other hand, iPSC differentiation into fully mature and functional liver cells still requires refinement. Liver organoid generation has been achieved with several protocols that involve different cell types at different stage of maturation. Liver organoids vascularized by the presence of endothelial cells in the cell mix show the best results in terms of survival and functionality when transplanted in mice models. Conversely, hepatoblast aggregates display biosynthetic and drug biotransformation properties, like the human liver, but their architecture is not maintained after transplantationTo overcome these problems, other systems generating organoids directly on microwell arrays advanced to methodologies using bioprinters have been applied. The 3D spheroid organization and liver specification has been improved; however, these novel systems are still complex and expensive and, to date, have never been transplanted in vivo to prove their feasibility.

## 8. Conclusions

Hepatic diseases are one of the major causes of death in the world and the costs for treatment will increase in the coming years [178]. Currently, the only treatment that is available is liver transplantation, but the lack of organs to be transplanted and the clinical instability of the patients make it difficult in reaching a cure for all patients. Alternative approaches have been explored, such as hepatocyte transplantation; however, this does not represent a definitive cure. To overcome these issues, several studies have focused on the development of mini liver-like structures, from both primary hepatic cells and iPSCs, which can replicate the composition of liver tissue offering a novel approach for disease modeling, drug screening, and regenerative medicine. Moreover, the use of iPSCs offers innumerable possibilities based on their ability to grow indefinitely in culture, to differentiate into almost all cell types, including both mature differentiated cells and tissue-specific stem cells, offering an alternative to the limited organ supply. In the development of liver diseases treatments, it is also important to replicate the physiological processes, such as vascularization, as well as cell-to-cell and cell-to-ECM interactions. Organoids exhibit exclusively, these relevant features. The 3D replication of human tissues offers the opportunity of better understanding the biological systems, thus making organoids a suitable tool for disease modeling. Several strategies have been developed to differentiate iPSCs into hepatocytes, cholangiocytes, endothelial cells, and KCs; however, difficulties have arisen in obtaining fully differentiated cells. Nevertheless, the construction and characterization of organoids have contributed to the maturation of iPSCs-derived liver cells making the picture of liver reconstruction much more feasible. Another advantage in the use of iPSCs for organoid generation is the possibility of using HLA-haplotype iPSCs to overcome the cost and the immunological limitation of iPSC-derived organoid cell therapy. Indeed, one of the major issues of liver transplantation remains the immunosuppressive therapy that HLA-haplotype iPSCs-derived organoids can potentially overcome. However, these cells cannot cover the entire variability present in the immunological population. Nevertheless, the use of universal cells can represent an alternative to HLA-haplotype iPSCs, overcoming the necessity to generate all possible haplotypes rendering them “invisible” to the immune system. However, this strategy still needs to be refined to limit the side effects related to the ability to escape immune surveillance. In recent years, the 3D technology has evolved from the generation of “liver-on-a-chip” to bio printing techniques. These systems could facilitate the maturation of iPSCs-derived liver cells and primary hepatic cells and fill the gap on the development of GMP-compliant organoids that are relevant to clinical applications. Overcoming the challenges in 3D-liver generation may lead to the replication of human physiological processes, confirming the capacity of organoids to adapt and respond over the long-term in the host microenvironment, allowing tissue engineering to become a reality for the cure of liver diseases.

## Figures and Tables

**Figure 1 ijms-21-06215-f001:**
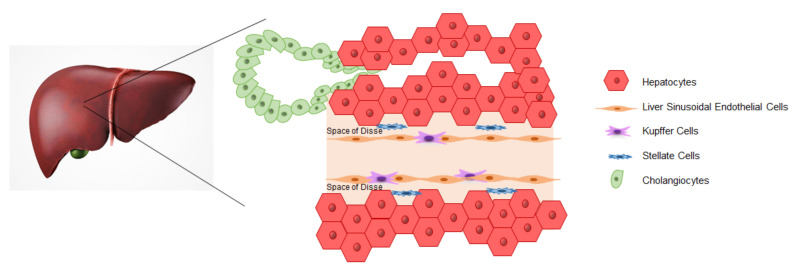
Schematic reproduction of the liver structure. Liver sinusoidal endothelial cells (LSECs) surround the hepatic sinusoids. The space of Disse separates hepatocytes and endothelial cells and contains the stellate cells. Hepatic macrophages (Kupffer cells) are in tight contact with LSECs and face the bloodstream. Cholangiocytes line the inner space of the bile duct tree.

**Figure 2 ijms-21-06215-f002:**
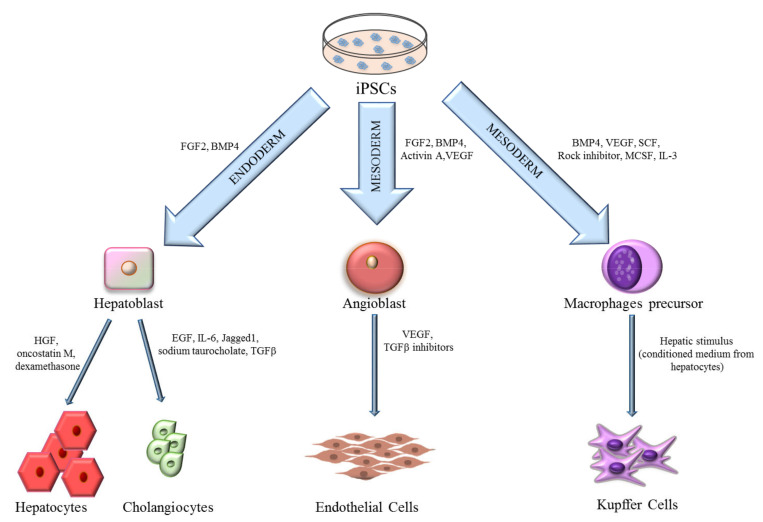
Differentiation methods of liver cells from induced pluripotent stem cells (iPSCs). iPSCs can be induced to mesoderm or endoderm by adding to the culture medium different cytokines. The differentiation process involves the precursor formation, specifically hepatoblast for hepatocytes and cholangiocytes, angioblast for endothelial cells and monocytes for Kupffer cells. FGF2, Fibroblast Growth Factor 2; BMP4, Bone Morphogenic Protein 4; VEGF, Vascular Endothelial Growth Factor; SCF, Stem Cell Factor; MCSF, Macrophage Colony-Stimulating Factor; IL3, Interleukin 3; HGF, Hepatocyte Growth Factor; EGF, Epidermal Growth Factor; IL6, Interleukin 6; TGFβ, Transforming Growth Factor β.

**Figure 3 ijms-21-06215-f003:**
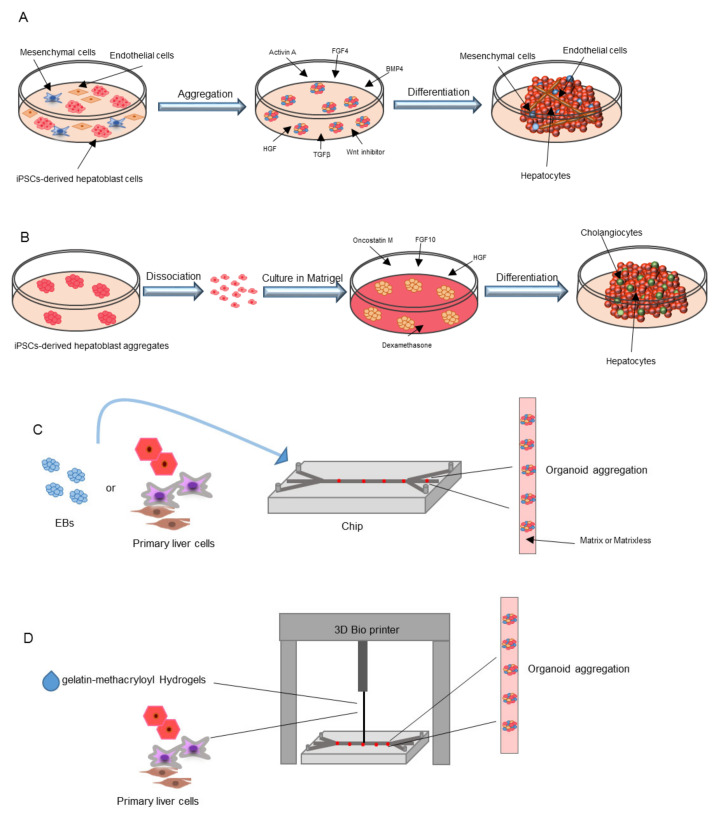
Methods for iPSCs-derived liver organoid generation. (**A**) Co-culture method by using endothelial cells, mesenchymal cells and iPSCs-derived hepatoblasts. After an initial aggregation step, several cytokines are added to the culture medium for organoid generation. (**B**) Organoids generated entirely from iPSCs derived cells. iPSCs-derived hepatoblast aggregates are dissociated, cultured in matrigel in presence of several cytokines and differentiate into liver organoids containing cholangiocytes and hepatocytes. (**C**) “Liver-on-a-chip” methods imply the culture of iPSCs-derived embryoid bodies or primary liver cells on chip in matrix or matrix independent conditions to induce organoid aggregation. (**D**) Three-dimensional (3D) printing technology implies the use of gelatin-methacryloyl Hydrogels as ink where primary hepatic cells are included and printed in transwells or perfused microwells.

**Table 1 ijms-21-06215-t001:** Summary of studies on the generation of iPSC-derived liver organoids.

Methodology	Author	Approach	In Vivo Transplantation and Survival	Advancement	Reference
**Co-Culture Method**	Takebe et al.	Co-culture of iPSCs-derived hepatic endoderm like cells with mesenchymal and human umbilical vein endothelial cells	Transplantation in TK-NOG mice after induced liver failure: survival up to 30 days	Vascularization of liver bud organoids and maturation in hepatocytes after in vivo transplantation	[113,114]
	Asai et al.	Co-culture of hepatic-specified endoderm iPSCs with mesenchymal and human umbilical vein endothelial cells	Implantation under the kidney capsule of immunodeficient mice: high serum levels of human albumin up to 8 weeks and hepatic maturation of the liver organoid in vivo	Paracrine factors secreted by mesenchymal cells and ECs (HGF, ANG, A2M, PLG) induce the formation of liver organoids	[121]
	Pettinato et al.	Co-culture of iPSCs-derived embryoid bodies with Human Adipose Microvascular Endothelial Cells (HAMEC)	An immune-deficient rat model for acute liver failure was transplanted with iPSCs-derived embryoid bodies + HAMEC: survival rate of 66.7% 14 days after induction of liver failure	The addition of HAMEC during hepatic differentiation of iPSCs induces liver-specific gene expression improving hepatic cell functionality	[125]
**iPSC Derived Organoids**	Koui et al.	iPSCs were differentiated into LSECs and hepatic stem cells by modulating TGFβ and Rho signaling pathways	ND	Self-renewal properties of iPSCs derived liver cells in 2D culture systems	[79]
	Wu et al.	iPSCs were differentiated into hepatobiliary organoids using Activin A, BMP4, BMP2, FGF4, HGF, OSM and dexamethasone	Hepatobiliary organoids were transplanted under the splenic capsule of immune-deficient mice: 4 weeks after transplantation biliary duct-like structures positive for human albumin were identified; 8 weeks after transplantation the hepatic structure was almost lost	By differentiating iPSCs into hepatobiliary organoids, there is no requirement of supportive cells, thus reducing costs and avoiding immune-rejection	[124]
	Guan et al.	iPSC-derived hepatic organoids containing both hepatocytes and cholangiocytes. CRISPR-Cas9 technology and piggyBac transposon system were combined to introduce and revert a mutation causing Alagille syndrome in healthy and patient derived iPSCs.	ND	The hepatic organoids generated showed biosynthetic and drug biotransformation properties similar to the human liver. The organoid after genome edited can be used for disease modeling and biology	[126]
	Ouchi et al.	First iPSCs were differentiated into foregut spheroids and then, in the presence of a hepatocyte specific medium, into a liver organoid containing hepatocytes, Kupffer, stellate and biliary cells	ND	The obtained organoids showed a transcriptomic profile comparable to hepatic tissue	[127]
	Toh et al., Jang et al., Zhu et al.	Hepatocytes or embryoid bodies were cultured in microfluidic 3D hepatocyte chip on collagen, Matrigel or hydrogel	ND	Matrices support the formation of 3D aggregates and can be used for drug testing	[128,129,130]
**Liver-on-a-Chip**	Norona et al., Nguyen et al., Bhise et al., Moya et al., Grix et al.	Gelatin-methacryloyl Hydrogels are used as ink. Primary hepatic cells were printed in transwell microwells to induce the generation of a 3D liver	ND	The coating of microfluidic chips supports the formation of 3D aggregates	[131,132,133,134,135]
**3D Printing Technology**	Goulart et al.	Bio compatible ink were used to print iPSC-derived parenchymal and non-parenchymal cells to generate a 3D liver organoid	ND	3D liver organoid showed hepatic functions	[136]

HGF, Hepatocyte Growth Factor; ANG, angiotensinogen; A2M, α-2 macroglobulin; PLG, plasminogen; LSECs, Liver sinusoidal endothelial cells; TGFβ, Transforming Growth Factor β; BMP4, Bone Morphogenetic Protein 4; BMP2, Bone Morphogenetic Protein 2; FGF4, Fibroblast Growth Factor 4; OSM, Oncostatin M.

**Table 2 ijms-21-06215-t002:** Summary of studies on liver organoid applications in regenerative medicine.

Author	Approach	Disease Mouse Model	Reconstitution	Follow Up	Reference
Huch et al.	Bile-duct derived organoids from Lgr5+ stem cells	Fumarylacetoacetate hydrolase (FAH)−/− mutant mice (model for Tyrosinemia type I liver disease)	0.1% of total liver volume	60–90 days	[112]
Huch et al.	EpCAM+ ductal cells from human liver biopsies induced to differentiate in hepatocytes and to form organoids	Balb/c nude mice treated with CCl4-retrorsine to induce acute liver damage	50–100 ng/mL of blood human albumin levels	120 days	[144]
Hu et al.	3D organoids from mouse and human primary hepatocytes	Fah−/− NOD Rag1−/− Il2rg−/− (FNRG) mice (model of tyrosinemia type I)	200 µg/mL on average of blood human albumin levels	90 days	[145]
Rashidi et al.	Organoid from iPSC-derived hepatocytes	Fumarylacetoacetate hydrolase (FAH)−/− mutant mice (model for Tyrosinemia type I liver disease) and Fah−/− NOD Rag1−/− Il2rg−/− (FNRG) mice (model tyrosinemia type I)	Detectable levels of human albumin	14 days	[146]
Blackford et al.	iPSC-derived hepatocytes generated with a cGMP compliant method was established to generate and seeded on a 3D poly-ethylene glycol-diacrylate scaffold to generate an organoid	Immune competent (C57BL/6 and Crl:CD1) and immune deficient (Rag2γ) mice	Detectable levels of human albumin	12 days	[147]

**Table 3 ijms-21-06215-t003:** Summary of studies on liver organoid applications as disease modeling.

Author	Approach	Disease model	Gene	Aim	Reference
Sampaziotis et al.	Human iPSCs from healthy donors and cystic fibrosis patients were differentiated into cholangiocyte-like cells	Cystic fibrosis associated biliary disease	Cystic fibrosis transmembrane conductance regulator gene (CFTR)	To test the effects of the drug VX809 on organoids	[51,157]
Guan et al.	iPSCs from healthy donors and Alagille syndrome patients were differentiated into 3D human hepatic organoids	Alagille syndrome	JAG1	To model Alagille syndrome. To introduce and revert JAG1 mutation with CRISPR/Cas9 technology and piggyBac transposon system.	[126]
Akbani et al.	iPSC-derived-EpCAM-positive endodermal cells differentiated into hepatic organoids	Citrullinemia type 1	Argininosuccinate synthetase (ASS1) gene	To model Citrullinemia type 1. To introduce the functional form of ASS1 by lentiviral vector transduction	[158]
Nie et al.	Human iPSC-derived endodermal, mesenchymal, and endothelial cells were cultured in specific medium to obtain liver organoids	Hepatitis B virus (HBV) infection	ND	To infect organoids from healthy iPSCs with HBV. To use infected organoids as a model of HBV infection.	[160]

**Table 4 ijms-21-06215-t004:** Summary of studies on liver organoid applications as liver tumors.

Starting Material	Author	Approach	Aim	Result	Limitation	Reference
**Needle Biopsies**	Takai et al.	Hepatocellular carcinoma cells cultured in porous alginate scaffolds generated spheroids	Mimic numerous features of glandular epithelium	Mimic numerous features of glandular epithelium	Tumor microenvironment interactions are not recapitulated	[165]
**PDX in Immunocompromised Mice**	Broutier et al.	PDX organoids from HCC	Drug sensitivity experiment	ERK inhibition could have an effect of HCC progression	Not suitable for immunotherapeutic approaches	[153]
	Gu et al.	PDX organoids from HCC	To generate a cohort of liver cancers to test a multi-kinase inhibitor	The drug was effective and used as a treatment for patients with advanced HCC	Not suitable for immunotherapeutic approaches	[167]
	Nie et al.	PDX organoids from HCC	To generate a cohort of liver cancers containing information about the expression profiles and the genetic alterations of all considered tumors	Identification of biomarkers for personalized medicine	Not suitable for immunotherapeutic approaches	[160]
	Saito et al.	PDX organoids from cholangiocarcinoma cells	To demonstrate that cholangiocarcinoma derives from differentiated hepatocytes	Restored hepatic functions	Not suitable for immunotherapeutic approaches	[171]
**PDX in Immunocompetent Mice**	Jiang et al.	HCC PDXs were generated in NSG gamma null mice repopulated with CAR-T cells	To study cancer immunotherapy	CAR-T cells directed against an HCC tumor-associated antigen suppressed tumor growth	Low engraftment of hematopoietic stem cells in the bone marrow of transplanted mice	[174]
	Choi et al.	HCCs generated in NSG mice with human leukocyte antigen-matched human immune systems	To study cancer immunotherapy	Organoids models were responsive to immunotherapies	Low engraftment of hematopoietic stem cells in the bone marrow of transplanted mice	[173]
**Organoids from Normal Tissues**	Artegiani et al.	healthy iPSCs or normal tissues	Introduce BAP1 and cholangiocarcinoma mutations by CRIPSR Cas9	Acquisition of malignant features	-	[177]

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
