# Peer review of "iPSC-Derived Liver Organoids: A Journey from Drug Screening, to Disease Modeling, Arriving to Regenerative Medicine"

_ijms, 2020, doi:10.3390/ijms21176215_

Round 1

Reviewer 1 Report

Well described manuscript, however, it does need some updates on the recent and useful papers to adapt as well as cite information from such papers.

  1. English grammar writing has to be improvised.
  2. In the introduction, the section has limited information on iPSC-derived hepatocytes and liver organoids ( even generally organoids).
  3. For figures 1 & 2, it is recommended to improve the quality by using professional software.
  4. It would be clear to reader if the iPSC-derived liver organoids section can be divided into different types of models. For example; next-gen liver models, vascularized models, cholangiocyte based organoids

Author Response

  • English grammar writing has to be improved
  • Response: we asked a native English speaking colleague to revise it and all the changes are in red troughs the text.

  • In the introduction, the section has limited information on iPSC-derived hepatocytes and liver organoids (even generally organoids)
  • Response: we added in the introduction a little section about liver organoids and iPSCs-derived hepatocytes. Lines 62-73

  • For figures 1 & 2, it is recommended to improve the quality by using professional software
  • Response: we improved the quality of the figures. We changed the original figures in the text with the one with high resolution.

  • It would be clear to the reader if the iPSC-derived liver organoids section can be divided into different types of models. For example; next-gen liver models, vascularized models, cholangiocytes based organoids
  • Response: we divided the iPSCs-derived liver organoids paragraph into several paragraphs that are: Co-culture Methods (Line 339), iPSCs-derived organoids (Line 368), Liver-on-a-chip (Line 380), Bioprinting technology (Line 392).

Reviewer 2 Report

The review by Follenzi et. al., titles iPSC-derived liver organoids: from drug screening, through disease modeling landing to regenerative medicine is of broad interest. This is important because of the rapid development in the field. It is a well-written review. I have the following comments:

  1. A simplified figure depicting the structure of the liver could help lay readers understand the goal of the organoid system.
  2. Section 5 (5. iPSCs-Derived Liver Organoids) requires a table that for example, categorizes a. approach, b. advancement, c. drawback and d. reference.
  3. Liver Organoids Applications section (6) also could use a table.
  4. In the disease modeling section, please include liver cancer and perhaps a table.
  5.  For the benefit of the readers, I think it would be a good idea to have a separate section that points out the challenges and room for advancements although most of which are already in the text.

Author Response

  • A simplified figure depicting the structure of the liver could help lay readers understand the goal of the organoid system
  • Response: we prepared a new figure representing liver structure (endothelial cells, hepatocytes, Kupffer cells and cholangiocytes) named figure 1. Thus, we changed the number of all the other figures.

  • Section 5 (5. iPSCs-Derived Liver Organoids) requires a table that for example, categorizes a. approach, b. advancement, c. drawback and d. reference
  • Response: we prepared a table for section 5, named Table 1, that categorize: author, approach, in vivo transplantation and survival, advancement and reference.

  • Liver Organoids Applications section (6) also could use a table
  • Response: we prepared two tables for section 6. The first (Table 2) is a summary of studies on liver organoids applications in regenerative medicine that categorize: author, approach, disease model, reconstitution, follow up and reference in the text. The second table for this section (Table 3) is a summary of studies on liver organoids applications in disease modeling that categorize: author, approach, disease model, gene, aim and reference.

  • In the disease modeling section, please include liver cancer and perhaps a table
  • Response: we added a new paragraph in liver organoids applications section named “Liver Cancer Organoids” (Lines 529-583) and a table for this part (Table 4) that categorize: author, approach, aim, results, limitations and reference.

  • For the benefit of the readers, I think it would be a good idea to have a separate section that points out the challenges and room for advancements although most of which are already in the text
  • Response: we added a new section named “Challenges and limitations of liver organoids” (section 7) Lines 584-601